# PiFM and XPS Studies of Porous TiO2 Films for the Photocatalytic Decomposition of Polystyrene

**Christopher Court-Wallace, Philip R. Davies \*, Josh Davies-Jones and Genevieve Ososki**

Cardiff Catalysis Institute, School of Chemistry, Cardiff University, Cardiff CF10 3AT, UK; daviesja21@cardiff.ac.uk
\* Correspondence: daviespr@cardiff.ac.uk

**Abstract:** The rate of photocatalytic oxidation of polystyrene over 0.1 wt% copper-doped TiO2 catalyst spin coated onto a flat substrate is investigated as a function of the catalyst deposition spin speed and, hence, film thickness. Photoinduced force microscopy and photoelectron spectroscopy show no evidence of any products of the photocatalytic oxidation remaining on the surface after reaction. The oxidation reaction shows no discernible dependence on spin speed; it is close to zero order in polystyrene concentration and exhibits a ½ life almost 50 times greater than the equivalent photocatalytic oxidation of stearic acid on the same catalysts. No difference between the rate of decay of the aliphatic and aromatic hydrogens of the polymer was observed, suggesting that once a polymer chain begins oxidising, subsequent steps are relatively rapid. This is consistent with the rate-determining step being dependent on the extent of coordination to the surface, which is much more favourable with stearic acid because of its carboxylic group.

**Keywords:** polystyrene; photocatalysis; surface; TiO2; spin coating

## 1. Introduction

The use of polystyrene is ubiquitous in today's society, most evidently as insulating hot drink containers and as a packaging material. Its very low weight, high stability, high thermal insulation and excellent shock absorption qualities have made it the polymer of choice across a very wide range of applications. In principle, polystyrene is also recyclable through mechanical granulation and re-incorporation into new polystyrene polymer or by compression into dense pallets. However, this polymer is not as widely recycled as others, its low-density, high-volume form creating a challenge for economic collection and transport. Polystyrene's chemical stability means that it has the potential to persist in the environment for thousands of years although recent studies suggest that its lifetime when exposed to sunlight is much shorter [1]. An early study by Achhammer et al. [2] reported the formation of hydroxyl and carbonyl groups at the polystyrene surface after exposure to 200 h of UV radiation in air at ~60 °C. Shang et al. studied the gas phase products evolved from polystyrene films photocatalytically degraded over embedded TiO2 and copper phthalocyanine modified TiO2 photocatalysts [3], reporting CO2, ethene, acetaldehyde, formaldehyde, and ethanol. More recently, Zan et al. [4,5] have studied TiO2 nanoparticles dispersed within a polystyrene film and chemically grafted to it through a silicone coupling agent. After exposure to UV, an increase in oxygen content in the surface was demonstrated at binding energies consistent with –CO2–, C–OH, and C=O groups. Support for these assignments is given by the development of intensity in the infrared spectra at 1721 cm⁻¹, characteristic of carbonyls. The authors also propose that oxidation occurs at the benzene rings based on a decreasing intensity in the characteristic phenyl ring vibrations at 1490, 1448, 750, and 701 cm⁻¹ with increasing exposure to light. However, Gardette, Mailhot, and Lemaire's [6] detailed study of direct photooxidation of polystyrene using FTIR and EPR only reported oxidation of the polymer backbone with the

phenyl rings being incorporated into products such as benzaldehyde and acetophenone rather than oxidised. The importance of effective methods for the degradation of polymers has been highlighted recently with the concern over the presence of nanoparticulates of polymers in water sources [7], and several investigations have looked at photocatalysis as a possible solution [8].

For our purposes, polystyrene is a useful model polymer for investigating decomposition strategies; its relatively simple mix of aromatic and aliphatic bonding provides a good comparison of their different degradation mechanisms. In the present study, we have been inspired by the reports of polystyrene breakdown under sunlight to examine the effectiveness of spin-coated titanium dioxide films for the photocatalytic decomposition of polystyrene films cast over the photocatalysts using the photocatalytic oxidation of stearic acid, which has been studied extensively in the past [9–11], as a reference point to compare photocatalytic activity. The work is part of a larger study of spin-coated photocatalysts [11] and concentrates on the effect of the deposition spin speed on the activity of the films towards the decomposition of the aromatic and aliphatic carbon-hydrogen bonds. We have selected a 0.1 wt% Cu-doped, porous $TiO_2$ film to test which shows significantly enhanced activity compared to undoped films in other reactions [11] and is, therefore, a promising choice to degrade the relatively stable polystyrene. Copper-doped $TiO_2$ catalysts are known to accelerate photocatalytic oxidation reactions, but there remains much debate on the mechanism [12,13]. In the catalysts studied here, we believe the copper to be evenly distributed as an interstitial dopant within the $TiO_2$ lattice but have not sought to investigate the enhancement mechanism. Instead, our aim was to consider the relative rates of aliphatic and aromatic C–H bond removal, to use photoinduced force microscopy (PiFM) and photoelectron spectroscopy (XPS) to look for any products of the reaction remaining on the surface after photocatalytic decomposition, and to compare the effect of spin coating speed (and, thereby, film thickness) on the catalytic activity. We show decomposition of the polystyrene to be independent of film thickness and almost 50 times slower than that of stearic acid with no intermediate products of the reaction detectable at the surface after reaction.

## 2. Results

### 2.1. Comparing the Photocatalytic Decomposition of Polystyrene with Stearic Acid

The photocatalytic oxidation of stearic acid to carbon dioxide and water is a well-established reaction for testing the photocatalytic activity of coatings [9,10]. Typically, the reaction is zero or first order [14] and leads to complete combustion of the deposited films. In our experiments, the photocatalytic decomposition of the stearic acid over a spin-coated porous 0.1 wt% $Cu/TiO_2$ layer proceeds with a half-life of approximately 250 s. XP and DRIFT spectra show no evidence for products left on the surface after complete photocatalytic decomposition. In contrast, the photocatalytic oxidation of polystyrene films is much slower, though still zero order, Figure 1, with a calculated half-life of ~12,000 s. Whereas stearic acid has only aliphatic carbons, polystyrene also has five aromatic C–H bonds for every three aliphatic, and these give rise to a slightly more complicated infrared spectrum. On the $Cu/TiO_2$ film deposited at 2000 rpm, the rate of decrease of the aliphatic hydrogen peak is very slightly greater than that of the aromatic; we return to this point in Section 2.3.

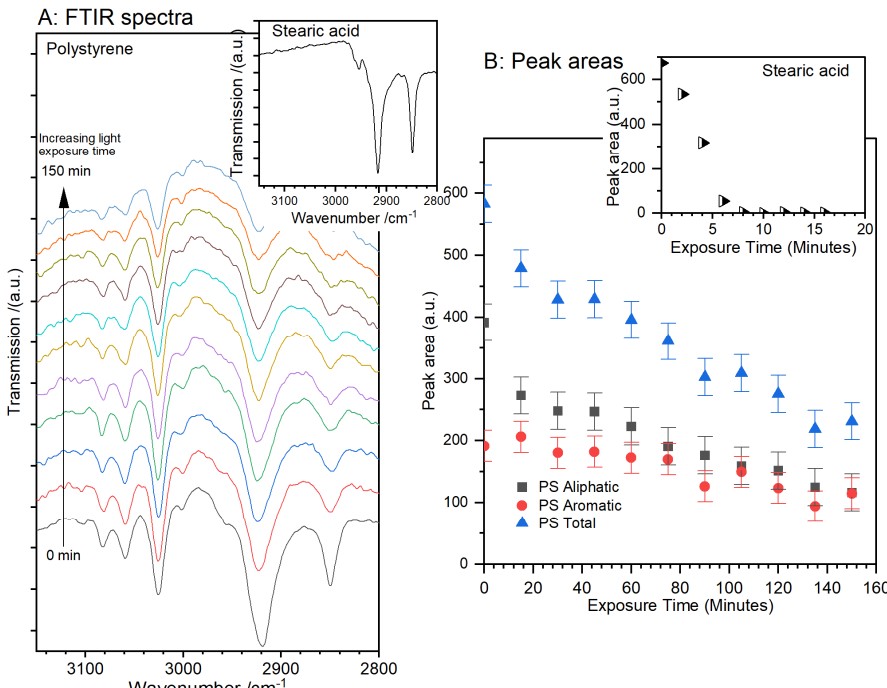

**Figure 1.** (**A**) Example experiment showing the effect of exposure to 365 nm light on the ν(C−H) intensity for a polystyrene film deposited on a porous Cu-doped TiO$_2$ film, spin coated at 2000 rpm; inset: a typical FTIR spectrum of stearic acid before photodecomposition. (**B**) Effect of irradiation on total ν(C−H) for polystyrene films with the contributions from the aliphatic and aromatic CH bonds shown separately. Inset: equivalent experiment with a stearic acid film. The calculated zero order ½ lives for stearic acid and polystyrene are ~250 s and 12,000 s, respectively.

### 2.2. Effect of Spin Speed and Photocatalytic Film Thickness on Activity for Polystyrene Decomposition

The photocatalytic decomposition of polystyrene was measured over a series of 0.1 wt% Cu/TiO$_2$ films deposited using spin speeds ranging from 1000 rpm to 8000 rpm, Figure 2A. There is no clear trend of activity with deposition spin speed (and, hence, film thickness), Figure 2B. The film deposited at 3000 rpm is a noticeable outlier with respect to polystyrene degradation, but none of our characterisations show any significant differences, and so we think this measurement can be disregarded. However, across all the samples, the rate of polystyrene decomposition is almost 50 times slower than that of stearic acid despite the aliphatic hydrogens present on both molecules.

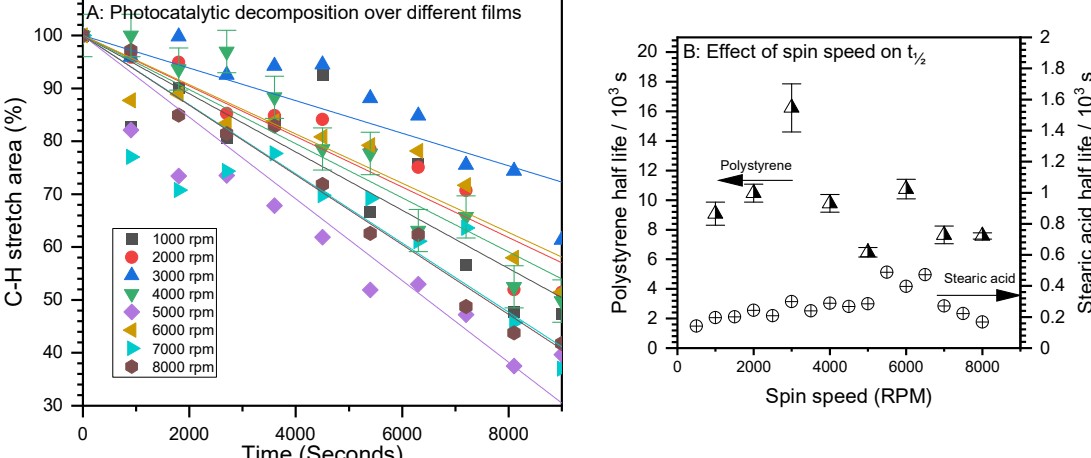

**Figure 2.** (**A**). Comparison of photocatalytic decomposition rates of polystyrene over Cu/TiO₂ films prepared at different spin rates. Error bars have only been included for one data set for clarity, but a typical error calculated from repetition is ±4%. (**B**): Calculated t½ lives of polystyrene vs spin speed compared with the t½ lives of stearic acid deposited on the equivalent films.

### 2.3. Comparing the Rate of Aliphatic and Aromatic Hydrogen Catalytic Oxidation

Whereas stearic acid only has aliphatic C−H groups, polystyrene offers the possibility of differentiating between the rate of catalytic oxidation of aromatic and aliphatic bonds. The porous TiO₂ films produced by spin coating at 2000 rpm appeared to show a slightly higher rate of oxidation of the aliphatic groups over the aromatic, and this evidence is confirmed through the comparison catalytic oxidation over films created with spin speeds between 1000 and 8000 rpm, Figure 3.

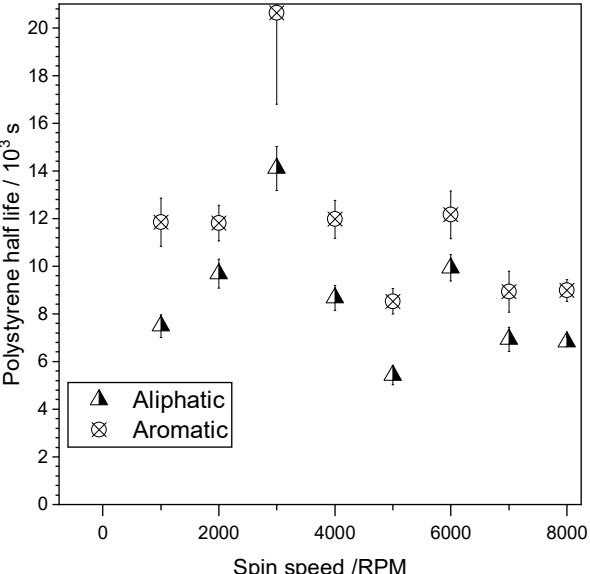

**Figure 3.** The effect of film deposition speed on the rate of photocatalytic decomposition of the aliphatic and aromatic C–H groups.

Whilst there is significant scatter in the results, every measurement shows a slightly longer ½ life for the aromatic over the aliphatic groups, but the difference is not as great as the difference in the overall rates of decomposition of the aliphatic only stearic acid and the mixed aliphatic/aromatic polystyrene.

## 2.4. Characterisation

XRD shows the spin-coated meso-porous TiO₂ films consist solely of the anatase polymorph, Figure S2, and, in contrast to previous reports of Cu-doped sol gel derived TiO₂, our data shows no effect of the copper doping as high as 5 wt% on the polymorph or on the dimensions of the TiO₂ lattice [15,16]. Tauc plots of the UV-VIS spectra also show no significant variation in the band gap of the films with copper concentration, Figure S3.

In the XP spectra, Figure 4 and Figure S5, the Cu 2p₃/₂ peak is only visible in the thicker films (slower deposition speeds) and where the concentration of copper is relatively high, but, in all cases, the Cu 2p peak appears at 932.7 eV, indicative of a Cu(I) or Cu(0) state [17], the former being the most likely although the usual satellite structure is too weak to give confirmation, and the LMM Auger peak is swamped by the Ti 2s peak. In the O(1s) region, the principal peak at 529.6 eV is due to TiO₂ and the higher binding energy component at ~532 eV to the SiO₂ substrate. A much smaller contribution at ~533 eV can be attributed to hydroxides or carbonates on the TiO₂ surface. In the C(1s) region, the main peak is at 284.8 eV with a smaller component at 288.5 eV that can be attributed to carbonates. The doublet at 292.8 and 295.5 eV is due to the K 2p signal from the glass substrate onto which the TiO₂ films are deposited. Deposition of the spin coated polystyrene, Figure 4, leads to the partial attenuation of all these peaks; the main carbon peak shifts to 284.5 eV, characteristic of a phenyl ring and accompanied by a weak shake-up satellite associated with delocalised systems at 291.2 eV [18]. Photocatalytic oxidation of the surface leads to the removal of the majority of carbon from the surface, with the main peak shifting back to 284.7 eV and the restoration of the intensity of all the other peaks.

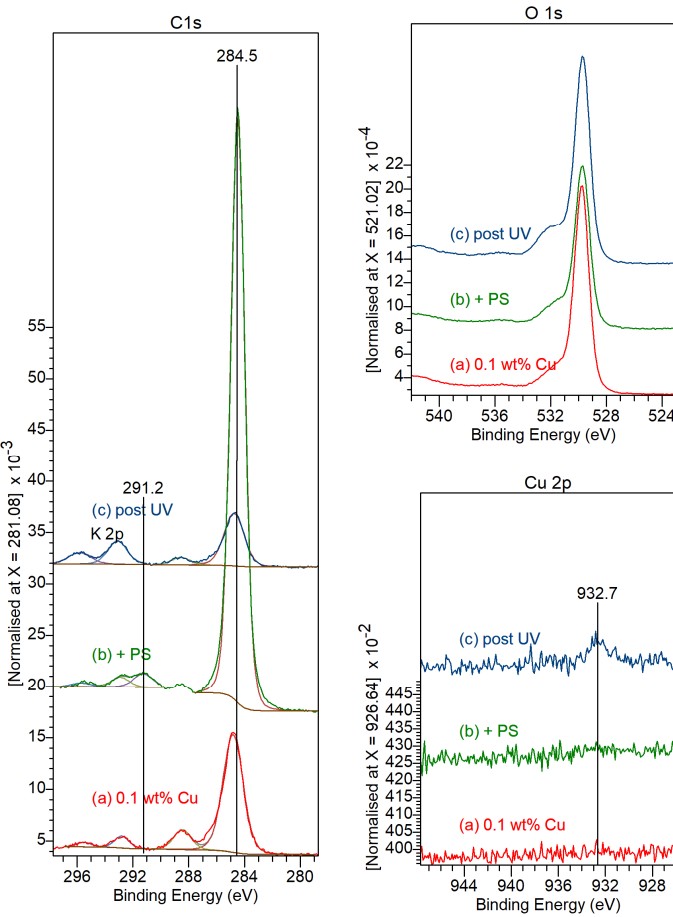

**Figure 4.** Photoelectron spectra of the 0.1 wt% Cu-doped TiO₂ catalysts. (**a**) Clean coating deposited at 500 rpm; (**b**) Coating deposited at 5000 rpm with polystyrene deposited on top. (**c**) Coating deposited at 5000 rpm with polystyrene after 72 h photocatalysis in air.

Unlike DRIFTS, photoinduced force microscopy, PiFM, provides infrared spectra of the surface region in the near field, with a lateral resolution < 10 nm; it also allows us to look at a wider wavenumber range [19]. We have examined the photocatalyst surface before and after exposure to UV light to look for the formation of decomposition products from the oxidation of the polystyrene on the surface, Figure 5. The polystyrene coated catalyst has height variation of ± 150 nm, and the deposited polystyrene is clearly identified by excellent correspondence between the PiFM spectrum and the FTIR transmission spectrum from a solid polystyrene sample, Figure 5a,b. After exposure to UV light, the topography of the surface is significantly rougher, increasing from RMS and Ra's of 85.17 and 71.49 nm, respectively, to 117.7 and 90.2 nm. The complete degradation of the polystyrene is evident from the dramatic change in colour between Figure 5e and Figure 5g; the blue colour of Figure 5e shows the high intensity of the characteristic 1495 cm$^{-1}$, whereas in Figure 5g the colour has switched to green, indicating 824 cm$^{-1}$, characteristic of the underlying TiO$_2$ film. Furthermore, there is no evidence for any other adsorbed species, including the in-phase ring CH bend at ~1030 cm$^{-1}$ expected [20] for any phenyl-ring-containing product. More detailed PiFM data, including height and intensity scales, are shown in Figure S6.

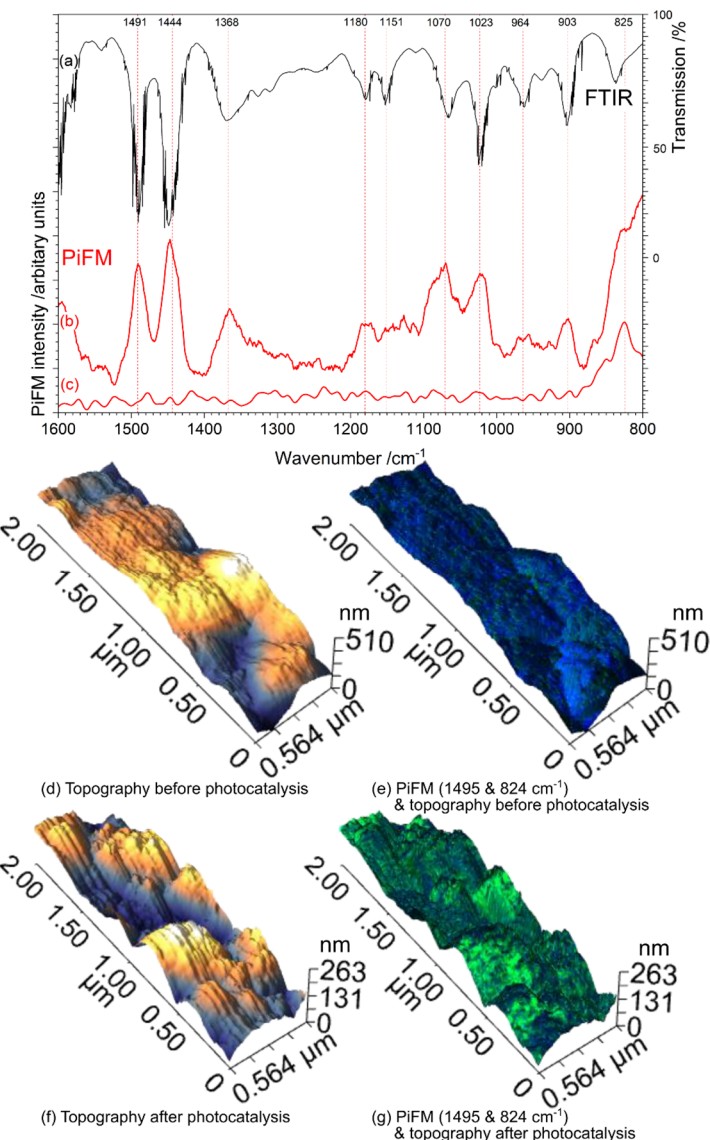

**Figure 5.** PIFM and AFM data on polystyrene covered TiO$_2$ films prepared at 5000 rpm. In the uppermost chart, the FTIR transmission spectrum from a polystyrene sample (**a**) is compared with the PiFM spectra obtained from the films before (**b**) and after photocatalysis (**c**). The topography of the

polystyrene covered catalyst surface before and after photocatalysis is compared in (**d**,**f**). In (**e**,**g**), the local intensities of the signals at 1495 cm$^{-1}$ (polystyrene, blue) and 824 cm$^{-1}$ (TiO$_2$, green) are mapped onto the topography.

## 3. Discussion

There is an urgent need to develop new approaches to break down plastics [1,21]; the question we are looking at here is how well a polymer such as polystyrene can be decomposed by the active oxygen species excited by light. The headline observation is that the rate of decomposition of polystyrene is more than 20 times slower than in the case of stearic acid. Interestingly, just as in the case of stearic acid, the PiFM and XPS indicate that there are no products remaining at the surface after photodegradation, suggesting complete oxidation to carbon dioxide and water. With no intermediates detected at any stage, we cannot speculate further about the mechanism of the decomposition, but a question we can consider is why the rate of oxidation of the polystyrene is so much slower than in the case of the stearic acid. It is unlikely that the decreased rate is due to the higher bond energy of the aromatic C–H groups, because, whilst this may contribute to the lower rate to some extent, our infrared data shows only a very small difference between the rate of removal of the aliphatic and aromatic hydrogens in the polystyrene. A reasonable hypothesis is that the much faster rate of oxidation of the stearic acid is related to the carboxylic headgroup. Previous work shows that an oxygenated group such as the carboxylic acid group in stearic acid, leads to a rapid oxidation, probably due to the initial covalent bonding of the molecule to the surface [22,23]. Zan et al. [4,5] chemically grafted TiO$_2$ nanoparticles to a polystyrene film through a silicone coupling agent in order to achieve rapid photocatalytic degradation. This close and strong surface bonding is missing in the case of polystyrene physisorbed at the surface; the critical step of interaction of the molecule with the active oxygen radicals is, therefore, less likely to occur. The PiFM data shows unequivocally that the photocatalytic oxidation process removes all products of the polystyrene from the surface.

It is very interesting that the different spin speeds used to deposit the copper-doped TiO$_2$ films on the silica substrates do not have a substantial effect on the rates of photocatalysis for either the stearic acid or the polystyrene despite the expectation that the resulting films have thicknesses ranging from 0.5 to ~3 μm and different morphologies (we are investigating these aspects in more detail for a future publication). The film thickness and physical structure would be expected to have an effect on the light absorption properties and rate of formation of hole/electron pairs at the surface, and so the relatively flat rate of reaction for the different conditions explored suggests that the rate-determining step is connected with the chemistry of the hydrocarbon/TiO$_2$ surface interaction.

## 4. Materials and Methods

### 4.1. Formation of 2D TiO$_2$ Spin Coated Films

Titanium (IV) butoxide (Ti(OBu)$_4$, Sigma-Aldrich reagent grade and copper (II) nitrate hydrate (Cu(NO$_3$)$_2$ × H$_2$O, Sigma Aldrich, Saint Louis, MO, USA) were used as precursors with methanol as a solvent. Pluronic P-123 (Sigma-Aldrich) was used as the templating polymer to achieve the mesoporous structure; 5 g P-123 was added to 15 cm$^3$ methanol (Sigma Aldrich, anhydrous 99.8%) and left to stir until the polymer was fully dissolved. Ti(OBu)$_4$ (10 cm$^3$) was added to the solution and left to stir for 30 min. Cu(NO$_3$)$_2$ was added, and the solution was left to stir for a further 20 min.

Cover glasses (VWR, 22 mm Ø, No. 1) were cleaned in water, then acetone using an ultrasonic bath at room temperature and placed in an oven at 40 °C. The titanium butoxide/ P-123/ Cu(NO$_3$)$_2$ solution was distributed onto the clean cover slips by spin coating with a Laurell (Lansdale, PA, USA), WS-650MZ-23NPP system. A total of 200 μL of the solution was dispensed onto a spinning cover glass; the coated cover glass was left to spin for a further 30 s. A range of spin speeds was used in order to achieve differing film thicknesses; for the present purposes, it is sufficient to report that faster deposition spin speeds

lead to thinner films and that the films are on the order of a micron thick. In a forthcoming publication [11], we examine the effect of spin speed on the thickness of these films in more detail. The samples were then heated to 40 °C for 48 h to facilitate the evaporation-induced self-assembly of the polymer and create a porous structure after which the samples were calcined in the furnace at 500 °C for 5 h with a heating rate of 5 °C min$^{-1}$ to remove the templating polymer.

### 4.2. Photocatalytic Testing

The photoactivity of the films was first measured with a standard stearic acid test [10] using diffuse reflectance infrared Fourier-transform spectroscopy (DRIFTS) on a Perkin Elmer (Waltham, MA, USA) Frontier spectrometer to monitor the surface concentration of C–H bonds of stearic acid. Illumination of the sample was conducted using an adapted UV-LED based photocatalytic test reactor [24,25] developed as part of the EU funded PCATDES project that provides a calibrated adjustable light source with a light intensity of up to 1.9 kW m$^{-2}$ at a wavelength of 365 ± 2 nm at the distance of 0.1 m from the UV-LEDs. The films were positioned c.a 10 cm from the LED source and photocatalysis conducted in air, and no temperature change was detected in the samples during photocatalysis.

A total of 200 μL of 0.1 mol dm$^{-3}$ stearic acid (Sigma Aldrich, reagent grade, 95%) in chloroform (reagent grade) was applied to the photocatalytic films using the spin coater at 2000 rpm for 30 s. Successive DRIFTS spectra were recorded before and after exposure to the LED lamp until the stearic acid had been completely decomposed or a time limit for the given sample had been reached. All measurements shown were an average of at least three experiments, and the narrow error range measured demonstrates the excellent reproducibility of the coating and reaction procedures. The percentage of stearic acid decomposed was calculated from the peak area under the C–H absorption peaks at 2917 and 2849 cm$^{-1}$ after subtraction of a constant background using OriginLab software 2022b, Supplementary Figure S1.

Decomposition of polystyrene was monitored in a similar way to the stearic acid. A 1 wt% solution of polystyrene (Sigma Aldrich, average Mw 35,000) in chloroform was prepared by weighing an individual pellet of polystyrene and dissolving in the appropriate amount of solvent, and then 200 μL of the solution was spin coated onto the photocatalyst at 2000 rpm. Analysis of these spectra required more complex fitting of the multiple absorptions present as there are both aliphatic and aromatic C–H stretches present, giving more information about the selectivity of the catalyst with respect to the two.

### 4.3. Catalyst Characterisation

XP spectra were obtained of the coatings with a Kratos Axis Ultra-DLD photoelectron spectrometer (Kratos Analytical, Manchester, UK) using a monochromatic aluminium K$\alpha$ x-ray source in the "hybrid spectroscopy" mode with an analysis area of 700 × 300 μm. A pass-energy of 40 eV was used for high-resolution scans and 60 eV for survey scans. CasaXPS [26] (version 2.3.24) was used to analyse the spectra with binding energies referenced to the largest Al$^{4+}$ (2p) peak at 458.5 eV with an uncertainty of ~0.2 eV. Raw spectra were modified using Wagner sensitivity factors, as supplied by the instrument manufacturer after the subtraction of a Shirley background.

Further characterisation was carried out on porous TiO$_2$ powders synthesised in the same way as above but not spin coated onto glass. Powder X-ray diffraction (XRD) data over the range 2θ = 10–80°, Figure S2, were obtained using a PANalytical (Malvern, UK) X'Pert Pro diffractometer with a monochromatic Cu K$\alpha$ source ($\lambda$ = 0.154 nm) operated at 40 kV and 40 mA. PiFM data were obtained using a VistaScope AFM platform manufactured by Molecular Vista (San Jose, CA, USA) exploiting a QCL laser (760–1860 cm$^{-1}$). The technique provides simultaneous topographic and vibrational spectroscopy information. In the present experiments, the sample was mapped at 1495 cm$^{-1}$ and 824 cm$^{-1}$ which are characteristic bands of the adsorbed polystyrene [19,27]. A surface area of 47.4 m$^2$ g$^{-1}$ was

measured for the powders using a Quantachrome NOVA 4200e instrument by $N_2$ adsorption using NovaWin v11.03 analysis software.

## 5. Conclusions

The rate of photodecomposition of polystyrene over a Cu promoted $TiO_2$ catalyst is 20× slower than the decomposition of stearic acid on the same catalysts.

The rates of removal of the aliphatic and aromatic hydrogens in the polystyrene are virtually equivalent.

Photodecomposition of polystyrene over the Cu–$TiO_2$ catalyst is complete, leaving no surface products.

The first use of Photo induced Force Microscopy to study photocatalysis has demonstrated the effectiveness of the method for providing local (ie nm) resolved information on the chemistry of the surface.

The relatively slow rate of polystyrene photodecomposition is attributed to the relatively weak interaction of the polymer with the surface. Whereas stearic acid chemisorbs strongly to the $TiO_2$ surface through the carboxylate functional group, polystyrene is only physisorbed. This conclusion is supported by the absence of evidence for any intermediate species at the catalyst surface, implying that the first step in the photodecomposition is the slow step, but reaction is very rapid once that initial barrier has been overcome. Interestingly, the rates of aliphatic and aromatic hydrogen removal are equal.

**Supplementary Materials:** The following supporting information can be downloaded at: https://www.mdpi.com/article/10.3390/catal13040725/s1, Figure S1: Comparison of ATR-FTIR spectra of spin-coated stearic acid and polystyrene on spin coated TiO2 films; Figure S2: Comparison of XRD of powder sample of undoped meso-porous TiO2 with copper-doped spin coated samples; Figure S3: Tauc Plots and calculated band gaps of the spin coated 0.1 wt% TiO2 catalysts; Figure S4: Photoelectron spectra of the Cu-doped TiO2 catalysts deposited at different spin speeds and with different wt% of Cu; Figure S5: Further photoelectron spectra of the 0.1 wt% Cu-doped TiO2 catalysts comparing the clean film with the polystyrene coated film before and after photocatalysis; Figure S6: PiFM micrographs of 0.1 wt% Cu-doped TiO2 treated with polystyrene before and after photocatalytic treatment.

**Author Contributions:** Conceptualization and methodology, P.R.D. and G.O.; formal analysis and investigation, G.O., C.C.-W., and J.D.-J.; funding acquisition, P.R.D. All authors have read and agreed to the published version of the manuscript.

**Funding:** XP spectra were acquired by EPSRC National Facility for Photoelectron Spectroscopy (HarwellXPS), operated by Cardiff University and UCL under contract number PR16195. The PiFM spectrometer was acquired with the EPSRC grant EP/V05399X/1. The European Regional Development Fund (ERDF) and the Welsh European Funding Office (WEFO) part-funded the Cardiff Catalysis Institute Microscopy facility. The light source used was provided by the European Union FP7 Project 309846, "Photocatalytic materials for the destruction of recalcitrant organic industrial waste (PCATDES)"

**Data Availability Statement:** The data presented in this study are openly available from Cardiff University at DOI: 10.17035/d.2023.0251718438.

**Acknowledgments:** EPSRC and the European Regional Development Fund (ERDF) for funding the equipment used in this study.

**Conflicts of Interest:** The authors declare no conflict of interest.

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
