# Peer review of "PiFM and XPS Studies of Porous TiO2 Films for the Photocatalytic Decomposition of Polystyrene"

_catalysts, doi:10.3390/catal13040725_

Round 1
Reviewer 1 Report
The article submitted for review examines the effectiveness of titanium dioxide films with a spin copper coating as a photocatalyst for the photocatalytic decomposition of polystyrene films.
The strengths of the reviewed article should include the style of presentation of information having a traditional structure, the brevity and clarity of the description of the methods used and the experimental dependencies obtained, the interpretation of the results obtained, as well as the quality of the graphic material.
ï€ On line 83 there is an inscription - "Error! Reference source not found.", apparently some information was displayed incorrectly.
ï€ What was meant on line 88 by the word "Section 0"?
ï€ There is no reference to Figure 1 in the text.
ï€ Why is the explanation to Figure 1 given in the caption to it (line 94, 95)?
ï€ Figure 1 (line 89) is followed by Figure 1 (line 106), that is, the numbering has gone astray.
ï€ The names of the parts of Figure 1 (line 89) and Figure 1 (line 106) are made in different styles.
ï€ Why, when discussing Figure 1b (line 106), the application speed of 5000 rpm is not considered, because it is clearly better for the decomposition of polystyrene?
ï€ In the text and in the figures, the abbreviation is used either rpm in capital letters, or RPM in capital letters.
Clarifying question: In Figure 3, as a pure coating, was the coating applied at 500 or at 5000 rpm considered?)
ï€ Subsection 4.1 describes the procedure for preparing a composition and applying a solution of Titanium (IV) butoxide, and compositions with a solution of Pluronic P-123, Titanium (IV) butoxide, copper (II) nitrate hydrate were applied in some other way?
Author Response
Comment: On line 83 there is an inscription - "Error! Reference source not found.", apparently some information was displayed incorrectly.
Response: This is a conversion problem from Word to PDF in the MDPI. We’ve fixed that by removing all the internal referencing.
Comment: What was meant on line 88 by the word "Section 0"?
Response: Similarly this is a conversion problem – now fixed
Comment: There is no reference to Figure 1 in the text.
Response: It is referred to in Line 83, but was hidden by the issue above.
Comment: Why is the explanation to Figure 1 given in the caption to it (line 94, 95)?
Response: This is the Figure legend. It is customary to describe the figure in the legend. Figure 1 is also described in the text. We think this is appropriate and have not changed the figure legend.
Comment: Figure 1 (line 89) is followed by Figure 1 (line 106), that is, the numbering has gone astray.
Response: Again this is the failure of the MDPI conversion to pdf to pick up the referencing in Word. We’ve taken out the internal referencing.
Comment: The names of the parts of Figure 1 (line 89) and Figure 1 (line 106) are made in different styles.
Response: In the document we submitted these are both in the style “MDPI_5.1_Figure.Caption”
Comment: Why, when discussing Figure 1b (line 106), the application speed of 5000 rpm is not considered, because it is clearly better for the decomposition of polystyrene?
Response: Although the measurements are within the calculated experimental error bars there is considerable scatter. There rate of reaction on the 5000 rpm films is not significantly different from the general rates to be identified as superior.
Comment: In the text and in the figures, the abbreviation is used either rpm in capital letters, or RPM in capital letters.
Response: Thank you. We have replaced the two instances of RPM with “rpm”
Clarifying question: In Figure 3, as a pure coating, was the coating applied at 500 or at 5000 rpm considered?)
Response: Similarly to the question above about the 500 rpm films. Neither the 500 rpm nor 5000 rpm deposited films are sufficiently different from the average to be deemed particularly good or bad.
Comment: Subsection 4.1 describes the procedure for preparing a composition and applying a solution of Titanium (IV) butoxide, and compositions with a solution of Pluronic P-123, Titanium (IV) butoxide, copper (II) nitrate hydrate were applied in some other way?
Response: Thank you for pointing out that discrepancy. The description is referring to the solutions with Cu(NO3) and pluronic acid present. The text in Section 4.1 has been clarified.
Reviewer 2 Report
Authors show quite comprechansive study of TiO2 porous films. The main focus is to study their synthesis in the activity aspect. Authors use various techniques FTIR / DRIFTS, XPS and microscopy to study all effects in the synthetic conntext. Important reaction - decomposition of plastics - is in the focus of current study. The topic is potentially innteresting for broad commuinty and has a value for the mdpi Catalysts.
I suggest to avoid abbreviations in the titel even if abbreviationns are obvious for readers.
Why experimental part is after discussion?
Line 83: reference is missing.
Fig 1B: vertical nunmbers are missing. If it is a.u. it makes less sense while there is nothing to compare.
There are Cu, C and O XPS edges discussed, I would also expect Ti edge to compare Ti state with Cu doping.
Short conclusions are missing. I would also expect more detailed discussion and comparison with other existinng catalysts.
Author Response
Thank you for your very helpful comments. We have been able to alter the manuscript for all of these comments follows:
Comment: I suggest to avoid abbreviations in the title even if abbreviationns are obvious for readers.
Response: Thank you, we thought about this a lot before suggesting the title because we agree that abbreviations are better avoided in titles. However, in this case the experimental techniques used are very important to the paper but their full names are far too long to put in the title. We therefore respectfully request that we retain the title as it stands.
Comment: Why experimental part is after discussion?
Response: This is the recommended structure from the MDPI Authors guidelines.
Comment: Line 83: reference is missing.
Response: Thank you, this is a problem with the MDPI Word/PDF converter we think. We’ve removed the internal references in the document which hopefully will solve the problem.
Comment: Fig 1B: vertical numbers are missing. If it is a.u. it makes less sense while there is nothing to compare.
Response: Thank you for pointing out that missing label, we’ve added it to the diagram. the axis label. We’ve also added the numbers to the y-axis on 1B.
Comment: There are Cu, C and O XPS edges discussed, I would also expect Ti edge to compare Ti state with Cu doping.
Response: The Ti2p peaks were recorded of course and they are reported in the Supplementary information: Figure S5. They don’t give us any useful information which is why they are not in the main paper.
Comment: Short conclusions are missing. I would also expect more detailed discussion and comparison with other existinng catalysts.
Response: Thank you, we have added a short conclusions section and included some discussion of other catalysts in the discussion section. However, we would emphasise that this paper is more about the behaviour of the catalyst and less about comparisons with other materials in the literature.
Round 2
Reviewer 2 Report
Authors addressed all comments and give sufficient improvements